# Integrated Bioinformatics Investigation of Novel Biomarkers of Uterine Leiomyosarcoma Diagnosis and Outcome

**DOI:** 10.3390/jpm13060985

**Published:** 2023-06-13

**Authors:** Aleksandar Rakic, Radomir Anicic, Marija Rakic, Lazar Nejkovic

**Affiliations:** 1The Obstetrics and Gynecology Clinic Narodni Front, 11000 Belgrade, Serbia; a.r.rakic@gmail.com (A.R.); radomir.anicic@gmail.com (R.A.); 2School of Medicine, University of Belgrade, 11000 Belgrade, Serbia; 3Faculty of Mathematics, Natural Sciences and Information Technologies, University of Primorska, 6000 Koper, Slovenia; mrakkic@gmail.com

**Keywords:** uterine leiomyosarcoma, bioinformatics analysis, *TYMS*, *TK1*, miR-26b-5p, Sp1

## Abstract

Uterine leiomyosarcomas (uLMS) have a poor prognosis and a high percentage of recurrent disease. Bioinformatics has become an integral element in rare cancer studies by overcoming the inability to collect a large enough study population. This study aimed to investigate and highlight crucial genes, pathways, miRNAs, and transcriptional factors (TF) on uLMS samples from five Gene Expression Omnibus datasets and The Cancer Genome Atlas Sarcoma study. Forty-one common differentially expressed genes (DEGs) were enriched and annotated by the DAVID software. With protein–protein interaction (PPI) network analysis, we selected ten hub genes that were validated with the TNMplotter web tool. We used the USCS Xena browser for survival analysis. We also predicted TF-gene and miRNA-gene regulatory networks along with potential drug molecules. *TYMS* and *TK1* correlated with overall survival in uLMS patients. Finally, our results propose further validation of hub genes (*TYMS* and *TK1*), miR-26b-5p, and Sp1 as biomarkers of pathogenesis, prognosis, and differentiation of uLMS. Regarding the aggressive behavior and poor prognosis of uLMS, with the lack of standard therapeutic regimens, in our opinion, the results of our study provide enough evidence for further investigation of the molecular basis of uLMS occurrence and its implication in the diagnosis and therapy of this rare gynecological malignancy.

## 1. Introduction

Even though they represent only 3–9% of all uterine malignancies, uterine leiomyosarcomas (uLMS) are aggressive neoplasms with poor prognosis, resistance to standard treatment protocols, and a high percentage of recurrent disease [1,2]. No imaging method or laboratory test can differentiate uLMS preoperatively with enough certainty [1,3]. The definitive diagnosis is still related to histology and, nowadays, molecular testing [4].

The main diagnostical challenge is to differentiate uLMS from uterine leiomyomas (ULM), a benign uterine smooth-muscle tumor with similar clinical and imaging findings. Since usual uterine leiomyoma treatment has a conservative or minimally invasive approach, the misdiagnosis between these entities could significantly increase morbidity and mortality [5,6].

Surgery and early complete resection is the only evidence-based effective treatment, with the standard protocol being total abdominal hysterectomy and bilateral salpingo-oophorectomy [7,8]. By the current consensus, chemotherapy is ineffective, and there are no standard regimens for adjuvant chemotherapy after complete resection [7,8,9,10]. On the other hand, doxorubicin monotherapy is still the superior regimen for metastatic, unresectable, and recurrent disease [8]. To date, none of the clinical trials performed a biomarker-specific patient selection [8].

Bioinformatics provides methodologies and databases for the analysis, integration, and interpretation of multi-omics Big Data [11]. Moreover, bioinformatics has become an integral element in rare cancer studies by overcoming the inability to collect a large enough study population [12]. With this study, we wanted to explore driving genes and significant pathways and identify potential novel biomarkers of diagnosis and outcome in uLMS patients using an integrated bioinformatics analysis.

## 2. Materials and Methods

### 2.1. Microarray Data Mining and Identification of DEGs

Supported by the National Center for Biotechnology Information (NCBI), Gene Expression Omnibus (GEO) (https://www.ncbi.nlm.nih.gov/geo/, accessed on 3 March 2023) is an open-access database that stores raw and processed gene expression data retrieved by a variety of methods, such as DNA microarray, high-through output sequencing, and RT-PCR [13,14]. We used advanced search queries to obtain datasets that include information regarding gene expression from uLMS, normal myometrium, and ULM. We included datasets that stored processed expression profiles obtained by array analysis of human uLMS, ULM, and normal myometrial tissue, with the number of samples > 8. Finally, we included the following datasets for further analysis: GSE764, GSE36610, GSE64763, GSE68312, and GSE32507. Information regarding the platforms, the total number of samples, and the specific number of tumor samples are presented in Table 1.

The Cancer Genome Atlas (TCGA) (https://www.cancer.gov/ccg/research/genome-sequencing/tcga, accessed on 4 March 2023), a joint project between the National Cancer Institute (NCI) and the National Human Genome Research Institute, is a publicly available storage of over 2.5 petabytes of genomic, epigenomic, transcriptomic, and proteomic data from over 20,000 cancer and matched normal tissue samples [15]. The TCGA-SARC dataset includes samples from several sarcoma subtypes, including leiomyosarcoma arising from gynecological tissue. The University of California, Santa Cruz (UCSC) Xena (https://xena.ucsc.edu/, accessed on 4 March 2023) is an online, freely available tool for visualizing and analysis of large public repositories and datasets, including TCGA [16]. The RNA sequencing (RNA-Seq) data, clinical data, and probe annotation files of 33 uLMS patients in TCGA-SARC were downloaded and analyzed by the UCSC Xena.

GEO includes an R-based tool for the analysis and visualization of differentially expressed genes between user-determined groups of samples from the exact GEO dataset, GEO2R (https://www.ncbi.nlm.nih.gov/geo/geo2r/, accessed on 5 March 2023) [17]. We used GEO2R to detect DEGs between uLMS and normal myometrium, uLMS and ULM, and uLMS and UCS samples. The screening of DEGS was carried out using a threshold of ∣log2 FC| ≥ 1 and *p* < 0.05. We used an online tool, InteractiVenn (http://www.interactivenn.net/, accessed on 10 March 2023) [18], to plot Venn diagrams of the DEGs of datasets. The overlapping DEGs from GSE764, GSE36610, GSE64763, and GSE68312 between uLMS and normal myometrium were enrolled in further analysis. DEGs between uLMS and uterine fibroids from GSE764 and GSE64763 were also screened and further analyzed. We screened DEGs between uLMS and UCS from GSE32507. Finally, we constructed Venn diagrams of the overlapped DEGs between two (uLMS-normal myometrium and uLMS-ULM) and all three cohorts, respectively.

### 2.2. Gene Ontology (GO) and Kyoto Encyclopedia of Genes and Genomes (KEGG) Pathway Analysis

GO is a bioinformatic resource that provides annotations supported by evidence to describe the biological roles of genes, proteins, and complexes, among others, by classifying them using predetermined ontologies [19]. The biological domain is explained by GO using three aspects: Molecular Function (MF), Cellular Component (CC), and Biological Process (BP). Kyoto Encyclopedia of Genes and Genomes (KEGG) integrates eighteen databases which are categorized into systems, genomic, chemical, and health information [20]. The central database in KEGG is PATHWAY, consisting of pathway maps. There are six categories of pathway maps: metabolism, genetic information processing, environmental information processing, cellular processes, organismal systems, and human diseases [20,21]. A comprehensive set of functional annotation tools is available in the Database for Annotation, Visualization, and Integrated Discovery (DAVID, https://david.ncifcrf.gov/home.jsp, accessed on 15 March 2023) to assist investigators in understanding the biological meaning behind large lists of genes. Using DAVID Gene as the foundation, these tools assemble functional annotations from multiple sources using the DAVID Knowledgebase. All DEGs were uploaded, separately analyzed, and visualized through DAVID software with a cutoff *p*-value < 0.05.

### 2.3. Protein-Protein Interaction (PPI) Network Construction

The STRING database (https://string-db.org/, accessed on 16 March 2023) is an online resource for the investigation of organism-wide protein association and interaction [22]. Each protein–protein association is accompanied by an online viewer allowing for visual inspection of the supporting evidence [22]. We used STRING version 11.5 with a cutoff confidence score > 0.4 for the construction of PPI between overlapped DEGs between uLMS and normal myometrium and uLMS and ULM.

### 2.4. Identification of Hub Genes

We uploaded and visualized the PPI network to Cytoscape version 3.9.1 software. We also used The Molecular Complex Detection (MCODE) [23], a Cytoscape plug-in, to contemplate the most significant nodes with degree cutoff = 2, K-Core = 2, and Node Score Cutoff = 0.2. Another Cytoscape plug-in, CytoHubba [24], was used to find the top genes with the Maximal Clique Centrality (MCC) analysis method, which has proven to be the superior method in predicting the essential proteins from the PPI network [24]. We also investigated the functional annotation and pathways enrichment of the selected genes. Finally, we constructed a gene co-expression network using GeneMANIA (https://genemania.org/, accessed on 20 March 2023) [25].

### 2.5. Validation of Hub Genes and Survival Analysis

Using TNMplot software (https://tnmplot.com/analysis/, accessed on 25 March 2023) [26], we compared the expression of selected hub genes between tumor and normal uterine tissue (both paired tumor and adjacent normal tissue and non-paired tumor and normal tissue platforms were used). To further verify the relationship between hub genes and clinical outcomes, we analyzed the data from the TCGA-SARC database for verification. Overall survival (OS) and disease-free interval (DFI) for the selected genes was performed using the USCS Xena browser.

### 2.6. TFs and miRNAs Related to Hub Genes

In order to map hub genes to their corresponding transcription factors (TFs) and miRNAs, we used NetworkAnalyst 3.0 (https://www.networkanalyst.ca/, accessed on 27 March 2023), a web-based visualization tool that facilitates the search for TF-gene and miRNA-gene interactions in gene regulatory networks [27]. miRNAs with a degree cutoff value = 1.0 were found for each of the hub genes.

We constructed TF-gene and miRNA-gene networks with Cytoscape. Furthermore, we obtained a table of the most significant TFs and miRNAs correlated with hub genes, ranked by an adjusted *p*-value, from miRTarBase (https://mirtarbase.cuhk.edu.cn/~miRTarBase/miRTarBase_2022/php/index.php, accessed on 30 March 2023) and TRRUST (https://www.grnpedia.org/trrust/, accessed on 30 March 2023), using Enrichr (https://maayanlab.cloud/Enrichr/, accessed on 30 March 2023) [28].

### 2.7. Drug–Hub Gene Interaction

We uploaded the hub genes to NetworkAnalyst 3.0 (https://www.networkanalyst.ca/, accessed on 10 April 2023) to obtain a drug-hub gene interaction network.

## 3. Results

### 3.1. Identification of DEGs

The analysis revealed 737 (266 upregulated and 471 downregulated), 1278 (504 upregulated and 774 downregulated), 1089 (392 upregulated and 697 downregulated), and 4145 (1986 upregulated and 2159 downregulated) DEGs between uLMS and normal myometrium, from GSE764, GSE36610, GSE64763, and GSE68312, respectively.

Furthermore, we found 57 overlapping DEGs (Figure 1A) from the four datasets (14 upregulated and 43 downregulated). There were 1102 (396 upregulated and 706 downregulated) and 752 (258 upregulated and 494 downregulated) DEGs between uLMS and ULM from GSE764 and GSE64763, respectively.

A total of 162 overlapped DEGs (Figure 1B) from these two databases were identified (32 upregulated and 130 downregulated). A volcano plot in Figure 1C shows 483 (290 upregulated and 193 downregulated) DEGs between uLMS and UCS from the GSE32507 dataset.

Figure 2A shows the intersected DEGs between uLMS and normal myometrium and uLMS and ULM. Two overlapping DEGs between all three cohorts are shown in Figure 2B. The only two overlapped DEGs between the three cohorts were *ATRX* and *PTGER3*, both downregulated.

### 3.2. GO and KEGG Pathway Enrichment Analysis

The most enriched GO terms and KEGG pathways for DEGs between uLMS and normal myometrium are presented in Table 2.

The most enriched GO terms and KEGG pathways for DEGs between uLMS and ULM are presented in Table 3.

Table 4 presents the most enriched GO terms and KEGG pathways for DEGs between uLMS and UCS.

The most enriched GO terms and KEGG pathways of the upregulated and downregulated overlapped DEGs between uLMS and normal myometrium and uLMS and ULM are presented in Table 5.

### 3.3. PPI Network and Hub Gene Selection

We uploaded overlapping DEGs between uLMS and normal myometrium and uLMS and ULM into the STRING database (Figure 3A). Then, the STRING data was uploaded to Cytoscape, where a PPI network with 30 nodes and 52 edges was constructed (Figure 3B). One module fulfilled the MCODE cutoff criteria (Figure 3C). This module consisted of: *TK1*, *TYMS*, *KIAA0101*, *CKS2*, *FOXM1*, *CCNE1*, *ESR1*, *MMP9*, *CXCL12*, *TGFBR2*, *CTGF*, and *IGF1*. Finally, the Cytohubba MCC module ranked the top 10 genes, which were classified as hub genes (Figure 3D, Table 6).

Among hub genes, there were six major upregulated (*TYMS*, *FOXM1*, *MMP9*, *CCNE1*, *CKS2*, and *TK1*) and four downregulated genes (*ESR1*, *CTGF*, *IGF1*, and *TGFBR2*). Figure 4 presents hub genes’ GO BP (Figure 4A), CC (Figure 4B), and MF (Figure 4C) annotations. Hub genes were enriched only in the “pathways in cancer” KEGG pathway. 

The present study utilized the GeneMANIA database to extract information regarding the interaction relationship and potential regulatory mechanism of the hub genes. Subsequently, a gene interaction network was constructed based on the obtained data. The network consists of 30 genes, including 10 hub genes and another 20 genes extracted from the GeneMANIA (Figure 5). The results displayed that the hub genes were co-expressed interactively with *BLZF1*, *MEP1A*, *CEP43*, *ZC3H11A*, *PKIG*, *FGFR3*, *MEP1B*, *VCAN*, *TGFBR3*, *E2F4*, *PNMT*, *COA7*, *PLK3*, *E2F3*, *CDK2*, *IGFBP6*, *IGFBP4*, *TGFB3*, *PES1*, and *IARS2*. The functions of the hub genes were mainly associated with growth factor binding, cell cycle G1/S phase transitions, transforming growth factor beta receptor (TGFBR) binding, and G1/S transition of the mitotic cell cycle (Figure 5).

### 3.4. Hub Gene Validation and Survival Analysis

The expression of overlapped DEGs between uLMS and normal myometrium and uLMS and ULM were validated using TNMplot. All upregulated overlapped DEGs were also overly expressed in tumor tissue compared to the normal uterine tissue (Figure 6). All downregulated overlapped DEGs except *ESR1* were significantly expressed in the normal uterine tissue compared to tumor tissue (Figure 7).

USCS Xena browsers’ advanced query isolated samples of uterine leiomyosarcoma with OS, DSS, and DFI data (Figure 8). Figure 9 shows the survival probability plots compared to the expression of previously identified hub genes. Since the analysis revealed that *TYMS* and *TK1* expression significantly correlated with overall survival, we further analyzed the DFI compared to the expression of these genes—lower *TK1* expression correlated with longer DFI. The correlation between *TYMS* expression and DFI was not significant (Figure 10).

### 3.5. Candidate TFs and miRNAs Related to Hub Genes

We constructed a miRNAs-target gene network with 397 nodes and 476 edges (Figure 11). Table 7 consists of six candidate miRNAs based on an adjusted *p*-value. miR-26a-5p and miR-26b-5p, two of the most significant miRNAs, were connected to five and seven hub genes, respectively. Candidate miRNAs targeting hub genes with adjusted *p*-values are presented in Table 7.

TF-hub gene network with 163 nodes and 227 edges is presented in Figure 12. Arguably the most significant TF, SP1, regulated seven hub genes (*TGFBR2*, *FOXM1*, *TYMS*, *CTGF*, *MMP9*, *TK1*, and *ESR1*). The rest of the TFs and overlapped genes are presented in Table 8.

### 3.6. Drug-Gene Interaction Network

Based on NetworkAnalyst, the only significant drug-gene interaction network was concentrated around TYMS. Figure 13 presents 20 potential drugs with which TYMS potentially interacts. Among them, we would highlight the interaction between TYMS and gemcitabine.

## 4. Discussion

With a low incidence, uLMS represents a rare gynecological malignancy. On the other hand, its aggressive behavior and molecular diversity urge studies and trials about the proper stratification of patients according to the molecular fingerprints to find the best therapeutic regimens for advanced and unresectable diseases. With an integrated bioinformatic approach, we wanted to explore the driver genes and relevant pathways of uLMS development, progression, and survival using the maximized available data. We used datasets with uLMS and healthy myometrial samples to investigate the potential genes included in malignant transformation. With the inclusion of DEGs between uLMS and ULM and uLMS and UCS, we wanted to highlight the genes that could potentially distinguish these entities preoperatively. Finally, GO and KEGG pathways enrichment, the construction of PPI to select the hub genes, validation and survival analysis of overlapped DEGs between uLMS, myometrium, and ULM could initiate a further investigation of highlighted genes as useful biomarkers of prognosis and therapy response.

DEGs between uLMS and myometrial tissue were enriched in the “response to estrogen” and “negative regulation of the apoptotic process” BP categories. The expression of estrogen receptors is present in approximately 50% of uLMS, with a range of 25–100% [29,30,31,32]. Estrogen receptor 1 (*ESR1*) was one of the most downregulated genes between uLMS and myometrium, as well as the downregulated hub gene in our study. TCGA study showed hypomethylation of *ESR1* response genes, which was one of the uLMS unique features compared to other sarcomas included in this study (mainly soft tissue sarcoma) [33]. Moreover, estrogen and progesterone receptors are considered biomarkers of prognosis in uLMS patients [30,34]. In correlation with rather high hormonal receptors expression in uLMS, there is evidence that uLMS shows a good response to aromatase inhibitors as an adjuvant therapy in newly-diagnosed patients with grade I uLMS, as well as those with recurrent, unresectable, and metastatic disease [31,35,36]. To date, only one clinical study showed that longer progression-free survival correlated with higher expression of the mentioned hormonal receptors [36].

One of the top upregulated DEGs across uLMS-normal myometrium and uLMS-ULM, also a hub gene in our study, was matrix metalloproteinase 9 (*MMP9*). *MMP9* is one of the most investigated members of this particular zinc-dependent endopeptidases family. The roles of *MMP9* occur as a result of extracellular matrix (ECM) degradation [37]. As a result of ECM and basement membrane degradation, it is believed that *MMP9* is involved in tumor migration, invasion, metastasis, and angiogenesis [38,39,40,41,42]. *MMP9* dysregulation is often associated with poor prognosis in ovarian, breast, and colon cancer patients [37]. *MMP9* is a well-studied biomarker in non-small cell lung, cervical, ovarian, and pancreatic cancer [37]. In our study, *MMP9*, a hub gene and upregulated DEG between uLMS-normal myometrium and uLMS-ULM cohorts, was part of the “negative regulation of apoptotic process” GO BP category, as well as “transcriptional misregulation in cancer” KEGG pathway enrichment. ESR1 and *MMP9* were also components of the only significant module of the PPI network. One study suggested that the change in *MMP* expression and cell motility, as a part of the decidualization process, could be estrogen-dependent and mediated by E2–ESR1–FOSL1 signaling pathway [43]. More studies should further investigate the relationship between *ESR1* and *MMP9* in the pathogenesis of uLMS, but it is possible that these two genes, alone or combined, have an influence on tumor behavior.

Besides *MMP9*, *FOXM1* (Forkhead Box M1) was a component of GO BP categories “negative regulation of the apoptotic process”, “positive regulation of cell proliferation”, and KEGG pathways “transcriptional misregulations in cancer”, and “pathways in cancer”, as well as major upregulated hub gene. *FOXM1* activates the expression of target genes at the transcriptional level, and the dysregulation of its activity can be observed in all hallmarks of tumor cells [44]. Substantial evidence highlights the role of *FOXM1* in cancer development: significant expression in a variety of human cancers [44], poor prognosis of most solid tumors with *FOXM1* overexpression [45], and the attenuation of angiogenesis, metastatic potential, and proliferation in some cancer types as a result of *FOXM1* inhibition [46].

*CKS2* (Cyclin-dependent kinase subunit 2) gene was upregulated hub gene in our study. *CKS2* is one of the two members of the human CKS family, believed to be an important factor in the process of somatic cell division during early embryonic development as well as the first metaphase/anaphase transition of meiosis [47,48]. Overexpression and upregulation of *CKS2* have been reported in breast [49], gastric [50], colorectal [51], and hepatocellular [52] cancer. Only one study to date investigated the role of *CKS2* in uLMS [53]. Deng et al. demonstrated significantly higher expression of *CKS2* in uLMS compared to ULM [53]. Furthermore, *CKS2* was associated with increased tumor size and poor overall prognosis in patients with uLMS [53]. Finally, silencing of *CKS2* inhibited cell proliferation, colony formation, migration, and invasion, and resulted in cell cycle arrest [53]. The authors hypothesize that *CKS2* may act as a cell cycle checkpoint protein for the G1/S transition [53]. These results, along with our findings, not only highlight the potential of *CKS2* as an excellent marker of differentiation between uLMS and ULM but provides enough evidence for the further investigation of this gene as a biomarker of prognosis and therapy response, as well as a novel therapy target in uLMS patients.

*ATRX*, *TK1*, and *TYMS* were essential genes in our study. In addition, *PTGER3* and *ARTX* were the only overlapped DEGs between the three cohorts, while *TK1* and *TYMS* were associated with the overall survival of the uLMS patients.

Studies recently highlighted the α-thalassemia/mental retardation syndrome X-linked (*ATRX*) gene as a central player in genome stability and function maintenance [54]. Gene expression, conservation of telomeric integrity, DNA damage repair, response to replication stress, and homologous recombination are processes in which *ATRX* has a crucial role [55,56,57]. Naturally, it is no coincidence that the *ATRX* is now one of the most studied tumor suppressors in a variety of human cancers. Telomeres, a non-coding sequence at each end of chromosomes, regulate chromosomal stability and genome integrity. As chromosome replication occurs in somatic cells, telomeres shorten with every duplication, since Okazaki fragments are synthesized with RNA primers attached ahead on the lagging strand, which leads to 3′ overhangs as the gap between the RNA primer and the end of the chromosome cannot be completed [58]. The majority of cells undergo programmed death when they encounter a barrier called “crisis” [58]. On the other hand, premalignant cells may overcome the crisis barrier by altering the telomere length pathway. These cells avoid the telomere shortening by two telomere maintenance mechanisms: telomerase-mediated telomere maintenance and alternative lengthening of telomeres (ALT) [59]. Emerging reports highlight the correlation between *ATRX* loss and the ALT process in human cancers [60]. In 2017, The Cancer Genome Atlas Research Network provided a multi-platform analysis of 206 different types of sarcomas, and they concluded that sarcomas are mostly characterized by copy-number changes, that they have relatively low mutational loads, and only a few genes highly mutated across all sarcoma types, one of them being *ATRX* [33]. A recent study found alterations of *ATRX* in 51% of uLMS [61]. Another gene mutated in 19% of uLMS, *DAXX*, is known to functionally cooperate with ATRX [61]. When Choi et al. investigated RNA levels between *ATRX*/*DAXX* mutation carriers and noncarriers, they confirmed that the detected alterations led to decreased gene expression [61]. This finding was also associated with ALT [61]. Finally, mutations in *ATRX* and TP53 correlated with poor prognosis in patients with uLMS [62].

Thymidine-kinase 1 (*TK1*) is involved in pyrimidine metabolisms and catalyzes gamma-phosphate group addition to thymidine [63]. *TK1* plays a significant role in the recovery pathways of pyrimidine nucleotide for DNA damage synthesis and repair [64]. *TK1* is also considered a valuable marker of cell proliferation, along with Ki-67 [63,64]. Some authors give an advantage to *TK1*, since its tight association with the S phase, while Ki-67 is present in all phases of the cell cycle [63,65]. So far, *TK1* has been applied as a biomarker in the lung [66], breast [64], prostate [63], and gastric cancer [67], and its overexpression was associated with poor prognosis. Furthermore, several studies identified *TK1*’s correlation with tumor aggressiveness [68,69,70]. A study by Wang et al. demonstrated the superiority of serum *TK1* over Carcinoembryonic antigen (CEA) and Alpha-fetoprotein (AFP) as a marker in the cancer screening of 56,286 people [71]. In this study, serum TK1 correlated with tumor growth rate and was also a prognostic biomarker for death at the follow-up [71]. Serum TK1 was more sensitive than CEA and AFP in discovering people with malignant tumors [71].

As far as we know, the role of *TK1* has yet to be studied in patients with uLMS. In our integrated bioinformatic analysis, *TK1* was the upregulated hub gene, and its lower expression correlated with overall and disease-free survival. We urge for the studies of *TK1* in uLMS since it could be a valuable biomarker for uLMS aggressiveness and a marker of the outcome in patients with uLMS.

*TYMS*, a gene that encodes thymidylate synthase, was upregulated hub gene in our study. Lower *TYMS* expression correlated with the overall survival of patients with uLMS. In DNA replication and repair, thymidylate synthase (TS) plays an essential role in the biosynthesis of thymidylate (dTTP) [72]. A study in vitro demonstrated that overexpression of *TYMS* causes immortalized mammalian cells to develop malignant phenotypes [73]. There is evidence that *TYMS* up-regulation is associated with adverse clinical behavior in a variety of solid tumor types, such as lung, breast, gastric, and colorectal cancer. In their bioinformatics study, Fu et al. demonstrated the upregulation of *TYMS* in pancreatic cancer and its association with poor overall survival and recurrence-free survival [74]. They concluded the study by proposing *TYMS* as a diagnostic and prognostic biomarker for patients with pancreatic cancer [74]. Zhang et al. demonstrated that the patients with higher *TYMS* expression had worse overall and disease-free survival in patients with retroperitoneal liposarcoma [75]. In this study, the knockdown of *TYMS* promoted apoptosis and reduced cell migration and invasion of retroperitoneal liposarcoma cells [75]. In a recent study, CRISPR-Cas9 functional telomere length screening revealed that thymidine nucleotide metabolism limits human telomere maintenance [76]. Targeted genetic disruption revealed several control points that included thymidine metabolism: deletion of the *TK1* decreased thymidine nucleotide salvage, while de novo knockout of *TYMS* decreased telomere length [76].

Specificity protein 1 (Sp1) is a transcriptional factor regulating the essential genes involved with cell proliferation and metastasis of various human neoplasms [77]. Patients with higher levels of Sp1 have a worse prognosis in several cancer types [77]. Dauer et al. found that the inhibition of Sp1 also causes cell death in pancreatic cancer [78]. Our results showed that Sp1 regulated seven hub genes. There is also a documented relationship between Sp1 and the upregulation of *MMP9*, a previously highlighted gene involved with tumor invasion and metastasis [79]. Several factors associated with telomere length regulation have been linked with Sp1 [79]. The relationship between Sp1, *TYMS* (which was found to be regulated by Sp1 in our study), and telomere length should be investigated in future studies.

In our study, miR-26a-5p and miR-26b-5p were the most significant miRNAs associated with hub genes. Both miRNAs were recently proposed as circulating biomarkers of several cancer types, including breast and cervical cancer [80,81]. The results of our study found an interesting link between *TYMS*, miR-26b-5p, and gemcitabine. First, we previously mentioned that *TYMS* expression correlated with OS in uLMS patients. Among seven hub genes, miR-26b-5p also targeted *TYMS*. Finally, our significant drug-gene interaction network highlighted the interaction between *TYMS* and gemcitabine. Adjuvant chemotherapy in uLMS patients remains controversial [82]. Gemcitabine is currently part of the Phase II trial as second-line chemotherapy for uLMS [8]. A multicenter study in Japan showed that the most frequent adjuvant chemotherapy was docetaxel and gemcitabine regimen [83]. A 2004 Phase II trial highlighted gemcitabine activity in uLMS patients and proposed its’ inclusion in multiagent regimens [84]. On the other hand, a recent study marked miR-26b-5p as one of the markers of gemcitabine resistance in patients with bladder cancer [85]. In addition to all these findings, the results of our study provide enough evidence for the future investigation of the role of *TYMS* in the pathogenesis of uLMS. Finally, *TYMS* could potentially serve as a valuable biomarker in uLMS prognosis.

There are several limitations to our study. Firstly, the conducted research via bioinformatic analysis does provide valuable results and novel insights into uLMS pathogenesis and prognosis, but these results require validation in the clinical setting. Furthermore, the expression of DEGs and hub genes should be verified via immunohistochemistry or even genetic studies and correlated with the clinical features of the uLMS patients. There are helpful, validated web tools for the verification of gene expression in various tissues and cancer types, such as The Human Protein Atlas (https://www.proteinatlas.org/, accessed on 1 May 2023). Unfortunately, there are no uLMS samples available for analysis. Finally, by integrating several datasets, we included a total of 90 uLMS patients. More uLMS samples are required to further validate our results.

## 5. Conclusions

Our integrated bioinformatic analysis identified several hub genes, candidate miRNAs, TFs, and signaling pathways associated with uLMS. Most importantly, TYMS, TK1, miR-26b-5p, and Sp1 panel should be further investigated as biomarkers of pathogenesis, prognosis, and differentiation of uLMS from other benign and malignant uterine tumors. Regarding the aggressive behavior and poor prognosis of uLMS, with the lack of standard therapeutic regimens, in our opinion, the results of our study provide enough evidence for studies to further investigate the molecular basis of uLMS occurrence and its implication in the diagnosis and therapy of this rare gynecological malignancy.

## Figures and Tables

**Figure 1 jpm-13-00985-f001:**
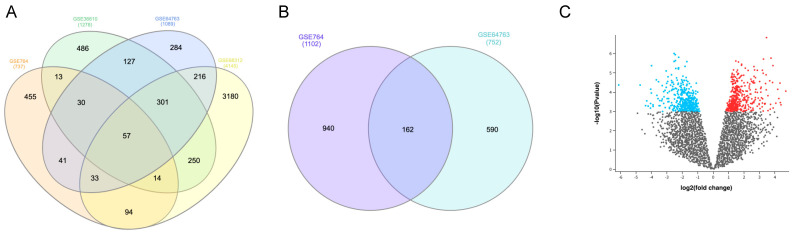
(**A**) Overlapping DEGs between uLMS and normal myometrium from GSE764, GSE36610, GSE64763, and GSE68312 datasets. (**B**) DEGs between uLMS and uterine leiomyomas from GSE764 and GSE64763 datasets. (**C**) Volcano plot with DEGs between uLMS and UCS from GSE32507 dataset.

**Figure 2 jpm-13-00985-f002:**
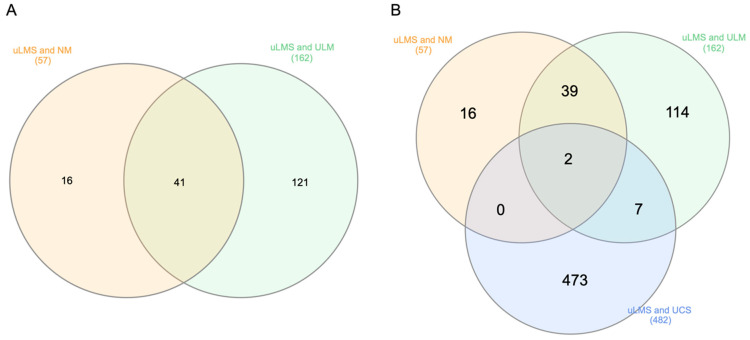
(**A**) Intersected DEGs between uLMS and normal myometrium and uLMS and uterine leiomyomas. (**B**) Overlapping DEGs between all three cohorts. NM—normal myometrium; ULM—uterine leiomyomas; UCS—uterine carcinosarcoma.

**Figure 3 jpm-13-00985-f003:**
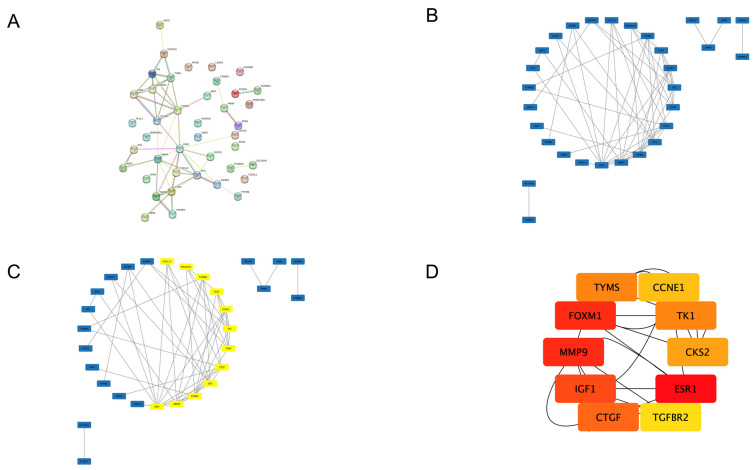
(**A**) PPI network constructed in STRING with overlapping DEGs between uLMS and ULM. (**B**) PPI network constructed by Cytoscape. (**C**) A significant module of the network (in yellow) provided by the MCODE plug-in. (**D**) Cytohubba MCC module ranked the hub genes from the following network (the color correlates with the rank: red rectangles present the highest rank, followed by orange and yellow).

**Figure 4 jpm-13-00985-f004:**
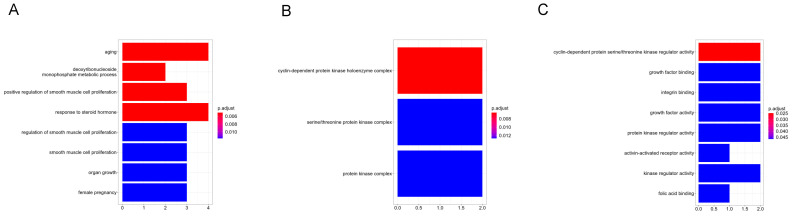
(**A**) GO BP annotation of hub genes. (**B**) GO CC annotation of hub genes. (**C**) GO MF annotation of hub genes. Color matching the adjusted *p*-value is presented on the right side of each category.

**Figure 5 jpm-13-00985-f005:**
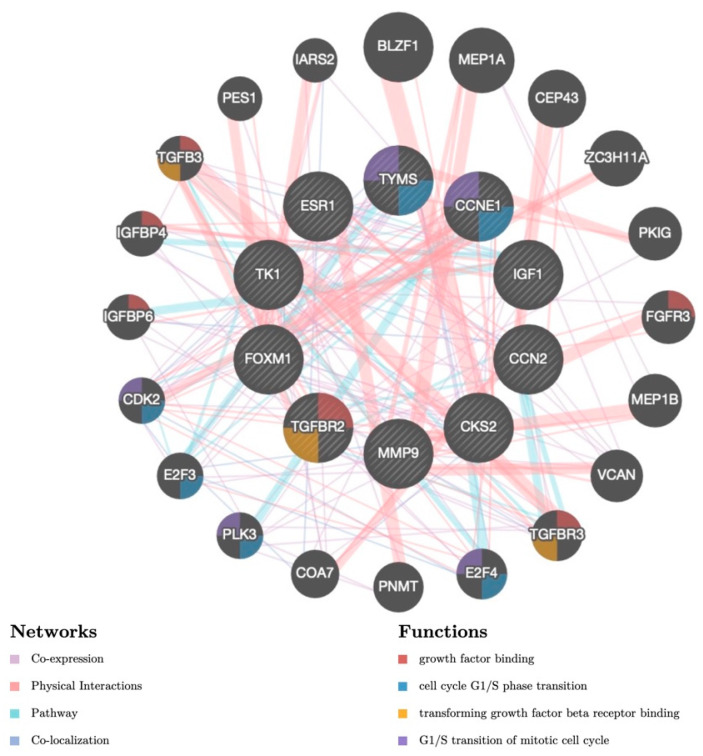
Gene co-expression network. Identified hub genes are presented in the inner circle; the outer circle is constructed of co-expressed genes. Interaction type and enriched functions of genes are presented with the corresponding colors below.

**Figure 6 jpm-13-00985-f006:**
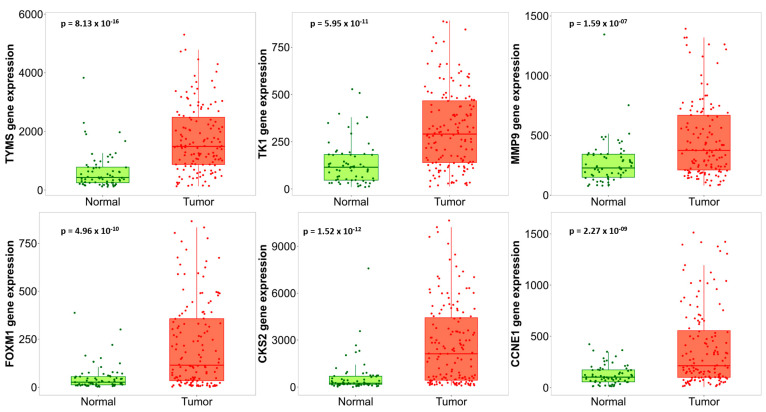
Expression of the upregulated hub genes in normal and tumor tissue.

**Figure 7 jpm-13-00985-f007:**
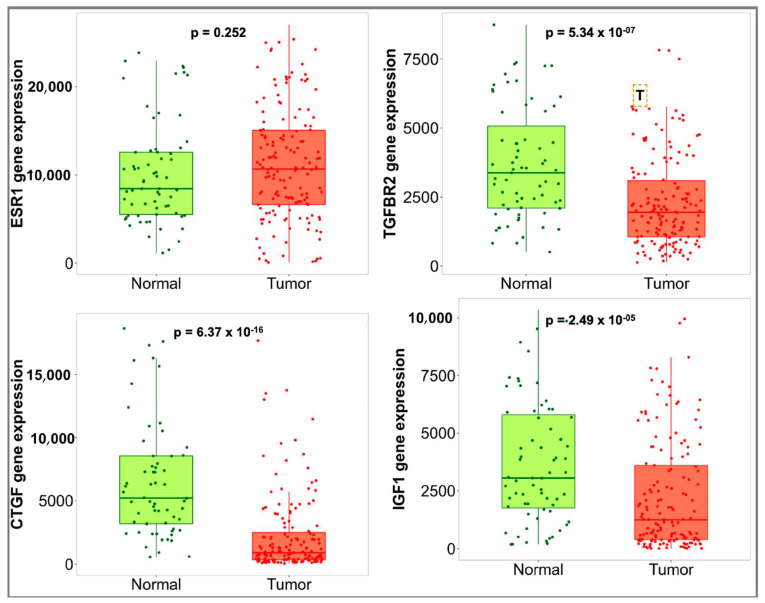
Expression of the downregulated hub genes in normal and tumor tissue.

**Figure 8 jpm-13-00985-f008:**
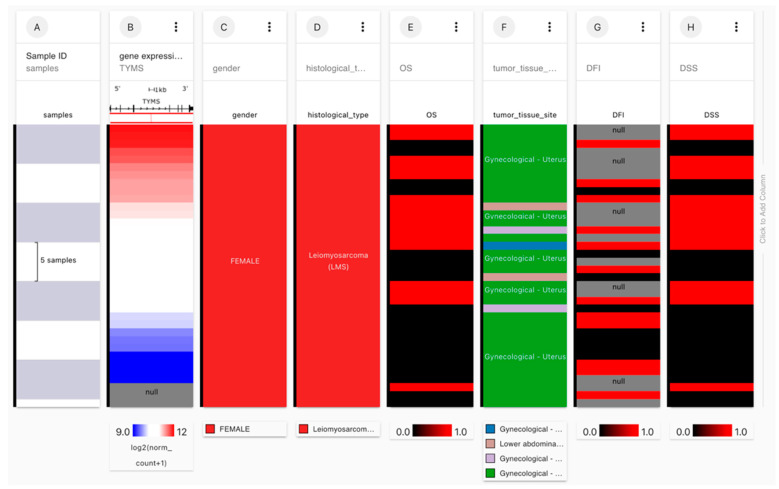
Extraction of uterine leiomyosarcoma samples from TCGA-SARC using USCS Xena. The samples were extracted from the TCGA-SARC study as follows: column A presents the number of samples; column B presents the expression of the gene of interest; gender: FEMALE (column C); histological type: Leiomyosarcoma (column D); tumor tissue site: Gynecological–Uterus (column F). Columns E, G, and H present Overall survival (OS), Disease-free interval (DFI), and Disease-specific survival (DSS), respectively. OS, DFI, and DSS were correlated with the expression of the genes of interest.

**Figure 9 jpm-13-00985-f009:**
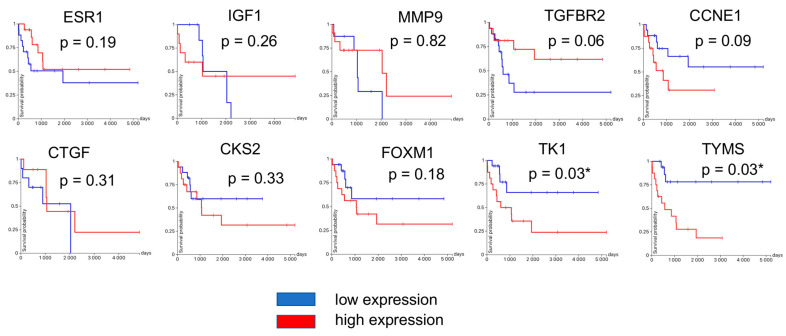
Overall survival of uLMS patients form TCGA-SARC compared to hub genes expression with *p*-values. *—presents a significant difference according to the *p*-value.

**Figure 10 jpm-13-00985-f010:**
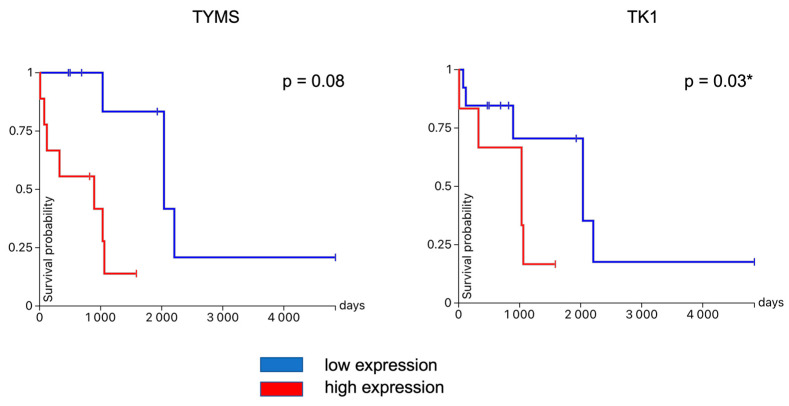
DFI of uLMS patients form TCGA-SARC compared to the expression of *TYMS* and *TK1*. *—presents a significant difference according to the *p*-value.

**Figure 11 jpm-13-00985-f011:**
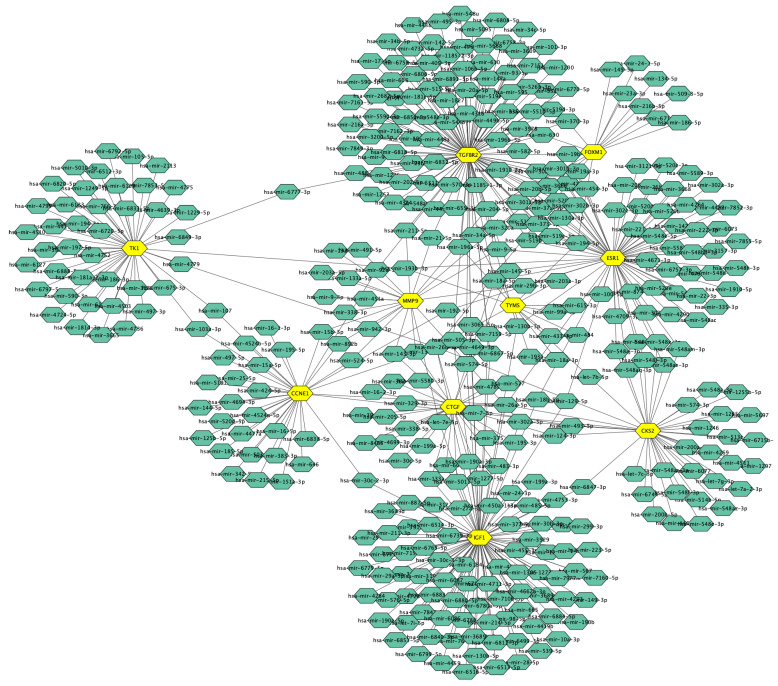
miRNA-target gene network. Yellow hexagons represent hub genes; green hexagons present miRNAs with a degree cutoff value of at least 1.0.

**Figure 12 jpm-13-00985-f012:**
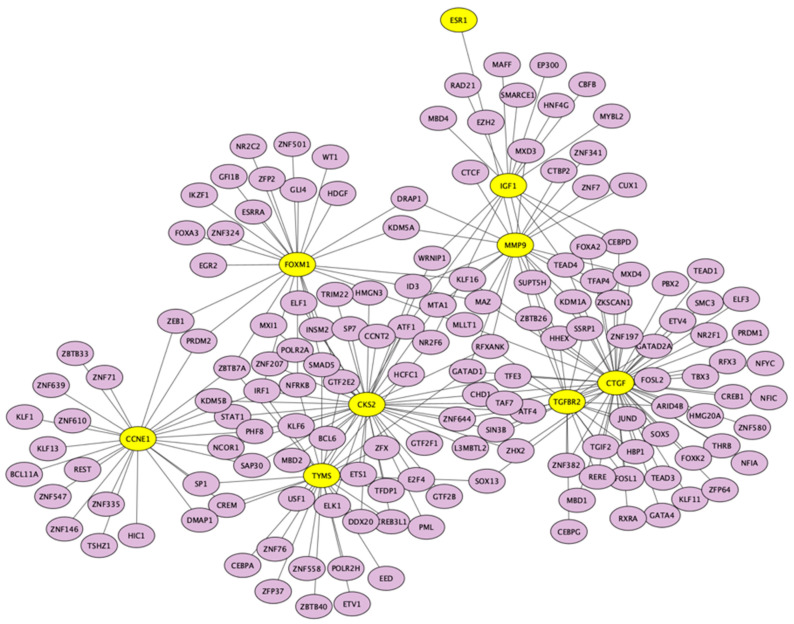
TF-hub genes network. Yellow circles represent hub genes; blue circles present corresponding TFs.

**Figure 13 jpm-13-00985-f013:**
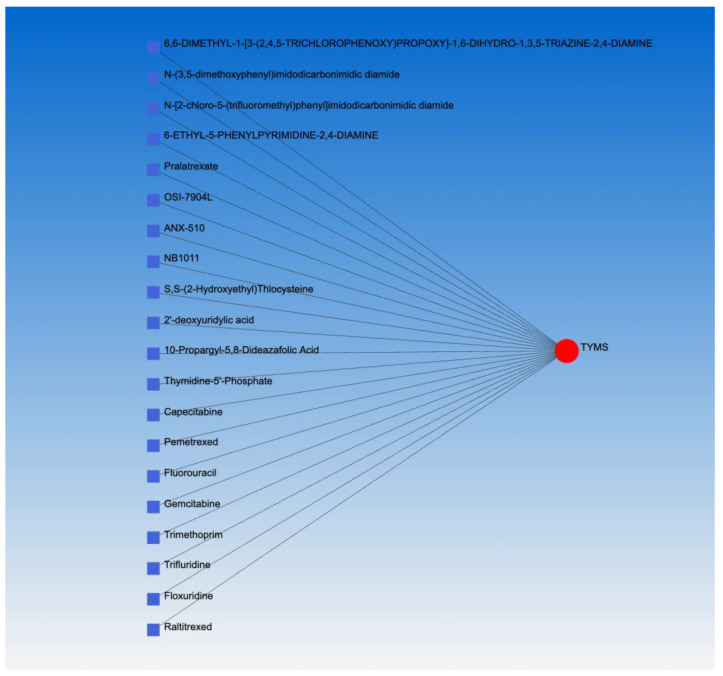
The only significant drug-gene interaction network based on NetworkAnalyst—twenty potential drugs (blue squares) reacting with TYMS (red circle).

**Table 1 jpm-13-00985-t001:** Platforms, total samples, and the number of tissue-specific samples from GSE datasets used for analysis.

GSE Series	Platform	No. of Samples	uLMS	Normal Myometrium	ULM	UCS
GSE764	GPL80	26	9	4	7	/
GSE36610	GPL7363	22	12	10	/	/
GSE64763	GPL571	79	25	29	25	/
GSE68312	GPL6480	9	3	3	3	
GSE32507	GPL6480	46	8	/	/	14

uLMS—uterine leiomyosarcoma; ULM—uterine leiomyoma; UCS—uterine carcinosarcoma.

**Table 2 jpm-13-00985-t002:** GO and KEGG pathway enrichment of DEGs between uLMS and normal myometrium with a number of genes enriched in selected terms and a *p*-value of significance.

Category	Term	Count	*p*-Value
GOTERM_BP_DIRECT	GO:0043627~response to estrogen	5	3.53 × 10^−5^
GOTERM_BP_DIRECT	GO:0043066~negative regulation of the apoptotic process	9	7.88 × 10^−5^
GOTERM_BP_DIRECT	GO:0007568~aging	6	1.75 × 10^−4^
GOTERM_BP_DIRECT	GO:0003151~outflow tract morphogenesis	4	2.91 × 10^−4^
GOTERM_BP_DIRECT	GO:0045944~positive regulation of transcription from RNA polymerase II promoter	12	3.09 × 10^−4^
GOTERM_CC_DIRECT	GO:0005887~integral component of plasma membrane	15	1.29 × 10^−5^
GOTERM_CC_DIRECT	GO:0005886~plasma membrane	28	1.16 × 10^−4^
GOTERM_CC_DIRECT	GO:0009897~external side of plasma membrane	8	2.41 × 10^−4^
GOTERM_CC_DIRECT	GO:0005576~extracellular region	16	2.77 × 10^−4^
GOTERM_CC_DIRECT	GO:0070062~extracellular exosome	15	1.39 × 10^−3^
GOTERM_MF_DIRECT	GO:0003682~chromatin binding	8	3.57 × 10^−4^
GOTERM_MF_DIRECT	GO:0042802~identical protein binding	14	8.97 × 10^−4^
GOTERM_MF_DIRECT	GO:0005539~glycosaminoglycan binding	3	2.49 × 10^−3^
GOTERM_MF_DIRECT	GO:0005319~lipid transporter activity	3	3.31 × 10^−3^
GOTERM_MF_DIRECT	GO:0005515~protein binding	46	5.18 × 10^−3^
KEGG_PATHWAY	hsa05202:Transcriptional misregulation in cancer	8	4.08 × 10^−5^
KEGG_PATHWAY	hsa05200:Pathways in cancer	10	1.01 × 10^−2^
KEGG_PATHWAY	hsa05215:Prostate cancer	4	1.23 × 10^−2^
KEGG_PATHWAY	hsa05205:Proteoglycans in cancer	5	1.86 × 10^−2^
KEGG_PATHWAY	hsa00240:Pyrimidine metabolism	3	3.38 × 10^−2^

**Table 3 jpm-13-00985-t003:** GO and KEGG pathway enrichment of DEGs between uLMS and ULM with a number of genes enriched in selected terms and a *p*-value of significance.

Category	Term	Count	*p*-Value
GOTERM_BP_DIRECT	GO:0008284~positive regulation of cell proliferation	26	2 × 10^−4^
GOTERM_BP_DIRECT	GO:0010628~positive regulation of gene expression	21	6 × 10^−6^
GOTERM_BP_DIRECT	GO:0007568~aging	12	4 × 10^−7^
GOTERM_BP_DIRECT	GO:0045944~positive regulation of transcription from RNA polymerase II promoter	28	1 × 10^−10^
GOTERM_BP_DIRECT	GO:0007179~transforming growth factor beta receptor signaling pathway	9	1 × 10^−10^
GOTERM_CC_DIRECT	GO:0005615~extracellular space	40	1.98 × 10^−7^
GOTERM_CC_DIRECT	GO:0005576~extracellular region	38	1.55 × 10^−10^
GOTERM_CC_DIRECT	GO:0005737~cytoplasm	72	2.06 × 10^−10^
GOTERM_CC_DIRECT	GO:0031093~platelet alpha granule lumen	7	1.31 × 10^−11^
GOTERM_CC_DIRECT	GO:0042383~sarcolemma	8	1.53 × 10^−11^
GOTERM_MF_DIRECT	GO:0005515~protein binding	134	1.19 × 10^−10^
GOTERM_MF_DIRECT	GO:0005158~insulin receptor binding	5	3.76 × 10^−10^
GOTERM_MF_DIRECT	GO:0005178~integrin binding	9	5.80 × 10^−10^
GOTERM_MF_DIRECT	GO:0005509~calcium ion binding	19	6.29 × 10^−10^
GOTERM_MF_DIRECT	GO:0005114~type II transforming growth factor beta receptor binding	4	6.70 × 10^−10^
KEGG_PATHWAY	hsa05205:Proteoglycans in cancer	15	5.27 × 10^−8^
KEGG_PATHWAY	hsa05200:Pathways in cancer	23	1.60 × 10^−10^
KEGG_PATHWAY	hsa05202:Transcriptional misregulation in cancer	14	1.62 × 10^−10^
KEGG_PATHWAY	hsa05206:MicroRNAs in cancer	15	6.25 × 10^−10^
KEGG_PATHWAY	hsa05218:Melanoma	7	3.86 × 10^−12^

**Table 4 jpm-13-00985-t004:** GO and KEGG pathway enrichment of DEGs between uLMS and UCS with a number of genes enriched in selected terms and a *p*-value of significance.

Category	Term	Count	*p*-Value
GOTERM_BP_DIRECT	GO:0097190~apoptotic signaling pathway	9	1.51 × 10^−4^
GOTERM_BP_DIRECT	GO:0043086~negative regulation of catalytic activity	10	9.65 × 10^−4^
GOTERM_BP_DIRECT	GO:0045214~sarcomere organization	6	1.09 × 10^−3^
GOTERM_BP_DIRECT	GO:0001933~negative regulation of protein phosphorylation	8	1.48 × 10^−3^
GOTERM_BP_DIRECT	GO:0051893~regulation of focal adhesion assembly	5	1.50 × 10^−3^
GOTERM_CC_DIRECT	GO:0005925~focal adhesion	30	1.24 × 10^−8^
GOTERM_CC_DIRECT	GO:0005829~cytosol	150	2.57 × 10^−11^
GOTERM_CC_DIRECT	GO:0070062~extracellular exosome	72	5.57 × 10^−10^
GOTERM_CC_DIRECT	GO:0005938~cell cortex	13	1.52 × 10^−11^
GOTERM_CC_DIRECT	GO:0016020~membrane	106	2.67 × 10^−11^
GOTERM_MF_DIRECT	GO:0005515~protein binding	327	1.28 × 10^−5^
GOTERM_MF_DIRECT	GO:0003779~actin binding	19	3.84 × 10^−11^
GOTERM_MF_DIRECT	GO:0002020~protease binding	10	6.18 × 10^−11^
GOTERM_MF_DIRECT	GO:0019901~protein kinase binding	23	1.00 × 10^−3^
GOTERM_MF_DIRECT	GO:0070513~death domain binding	3	4.00 × 10^−3^
KEGG_PATHWAY	hsa04510:Focal adhesion	19	4.01 × 10^−7^
KEGG_PATHWAY	hsa04810:Regulation of actin cytoskeleton	15	5.22 × 10^−4^
KEGG_PATHWAY	hsa04270:Vascular smooth muscle contraction	11	7.31 × 10^−4^
KEGG_PATHWAY	hsa05135:Yersinia infection	10	3.22 × 10^−3^
KEGG_PATHWAY	hsa05418:Fluid shear stress and atherosclerosis	10	3.55 × 10^−3^

**Table 5 jpm-13-00985-t005:** GO and KEGG pathway enrichment of the upregulated and downregulated overlapped DEGs between uLMS and normal myometrium and uLMS and ULM with a number of genes enriched in selected terms and a *p*-value of significance.

UPREGULATED	**Category**	**Term**	**Count**	***p*-Value**
	GOTERM_BP_DIRECT	GO:0051726~regulation of cell cycle	3	5.02 × 10^−3^
	GOTERM_BP_DIRECT	GO:0044772~mitotic cell cycle phase transition	2	9.88 × 10^−3^
	GOTERM_BP_DIRECT	GO:0071897~DNA biosynthetic process	2	1.36 × 10^−2^
	GOTERM_CC_DIRECT	GO:0000307~cyclin-dependent protein kinase holoenzyme complex	2	1.66 × 10^−2^
	GOTERM_MF_DIRECT	GO:0019901~protein kinase binding	3	1.87 × 10^−2^
	KEGG_PATHWAY	hsa00240:Pyrimidine metabolism	2	4.17 × 10^−2^
	KEGG_PATHWAY	hsa05200:Pathways in cancer	3	5.27 × 10^−2^
	KEGG_PATHWAY	hsa01232:Nucleotide metabolism	2	6.06 × 10^−2^
DOWNREGULATED	GOTERM_BP_DIRECT	GO:0007568~aging	5	7.45 × 10^−5^
	GOTERM_BP_DIRECT	GO:0045944~positive regulation of transcription from RNA polymerase II promoter	9	2.9 × 10^−4^
	GOTERM_BP_DIRECT	GO:0043066~negative regulation of the apoptotic process	6	9.56 × 10^−4^
	GOTERM_CC_DIRECT	GO:0005886~plasma membrane	19	1.49 × 10^−4^
	GOTERM_CC_DIRECT	GO:0005887~integral component of plasma membrane	9	1.03 × 10^−3^
	GOTERM_CC_DIRECT	GO:0005576~extracellular region	10	3.14 × 10^−3^
	GOTERM_MF_DIRECT	GO:0005539~glycosaminoglycan binding	3	8.25 × 10^−4^
	GOTERM_MF_DIRECT	GO:0003682~chromatin binding	5	6.81 × 10^−3^
	GOTERM_MF_DIRECT	GO:0001228~transcriptional activator activity, RNA polymerase II transcription regulatory region sequence-specific binding	5	7.28 × 10^−3^
	KEGG_PATHWAY	hsa05202:Transcriptional misregulation in cancer	7	8.54 × 10^−6^
	KEGG_PATHWAY	hsa05200:Pathways in cancer	7	2.18 × 10^−3^
	KEGG_PATHWAY	hsa04068:FoxO signaling pathway	3	4.75 × 10^−2^

**Table 6 jpm-13-00985-t006:** Hub genes ranked by MCC method from Cytohubba plug-in, with the full gene names, regulation (downregulated/upregulated), and MCC score.

Gene Symbol	Gene Name	Regulation	Score
*ESR1*	estrogen receptor 1	downregulated	60
*FOXM1*	forkhead box M1	upregulated	55
*MMP9*	matrix metallopeptidase 9	upregulated	55
*IGF1*	insulin-like growth factor 1	downregulated	52
*CTGF*	connective tissue growth factor	downregulated	51
*TK1*	thymidine kinase 1	upregulated	50
*TYMS*	thymidylate synthetase	upregulated	50
*CKS2*	cyclin-dependent kinases regulatory subunit 2	upregulated	48
*CCNE1*	cyclin E1	upregulated	30
*TGFBR2*	transforming growth factor, beta receptor II	downregulated	26

**Table 7 jpm-13-00985-t007:** Candidate miRNAs targeting hub genes with the number of overlapped genes, adjusted *p*-values.

Term	Overlap	Adjusted *p*-Value	Genes
hsa-miR-26b-5p	7/1872	0.0012	*CCNE1*; *CKS2*; *IGF1*; *TK1*; *FOXM1*; *TYMS*; *CTGF*
hsa-miR-18b-5p	3/116	0.0029	*IGF1*; *ESR1*; *CTGF*
hsa-miR-302a-5p	3/126	0.0029	*CKS2*; *IGF1*; *MMP9*
hsa-miR-145-5p	3/238	0.015	*ESR1*; *CTGF*; *TGFBR2*
hsa-miR-18a-5p	3/262	0.017	*ESR1*; *CTGF*; *TGFBR2*

**Table 8 jpm-13-00985-t008:** List of the most significant TFs, number of overlapped genes, ranked by *p*- and q-values, and list of overlapped genes.

Key TF	Description	No. of Overlapped Genes	*p*-Value	*q*-Value	List of Overlapped Genes
SP1	Sp1 transcription factor	7	6.57 × 10^−10^	1.45 × 10^−8^	*TGFBR2*, *FOXM1*, *TYMS*, *CTGF*, *MMP9, TK1, ESR1*
FLI1	Friend leukemia virus integration 1	3	2.16 × 10^−7^	2.28 × 10^−6^	*TGFBR2*, *FOXM1*, *CTGF*
HDAC2	Histone deacetylase 2	3	3.11 × 10^−7^	2.28 × 10^−6^	*CCNE1*, *TGFBR2*, *IGF1*
EP300	E1A binding protein p300	3	2.93 × 10^−6^	1.36 × 10^−5^	*IGF1*, *CCNE1*, *MMP9*
WT1	Wilms tumor 1	3	3.09 × 10^−6^	1.36 × 10^−5^	*IGF1*, *CCNE1*, *CTGF*
NCOR1	Nuclear receptor corepressor 1	2	5.3 × 10^−6^	1.94 × 10^−5^	*IGF1*, *ESR1*
STAT5B	Signal transducer and activator of transcription 5B	2	7.07 × 10^−6^	2.22 × 10^−5^	*IGF1*, *ESR1*
ETS1	V-ets erythroblastosis virus E26 oncogene homolog 1 (avian)	3	8.3 × 10^−6^	2.28 × 10^−5^	*TGFBR2*, *CTGF*, *MMP9*
TFDP1	Transcription factor Dp-1	2	1.14 × 10^−5^	2.77 × 10^−5^	*CCNE1*, *TYMS*

## Data Availability

Microarray data from uterine leiomyosarcomas, normal myometrium, uterine leiomyomas, and uterine carcinosarcomas are available in Gene expression omnibus datasets (https://www.ncbi.nlm.nih.gov/geo/, accessed on 3 March 2023). Survival data from uterine leiomyosarcoma patients are available in The Cancer Genome Atlas Sarcoma (TCGA-SARC) study (https://portal.gdc.cancer.gov/projects/TCGA-SARC, accessed on 4 March 2023).

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
