# Peer review of "Integrated Bioinformatics Investigation of Novel Biomarkers of Uterine Leiomyosarcoma Diagnosis and Outcome"

_jpm, 2023, doi:10.3390/jpm13060985_

Round 1
Reviewer 1 Report
Dear Editor, thank you for letting me review this manuscript entitled “Integrated bioinformatics investigation of novel biomarkers of uterine leiomyosarcoma diagnosis and outcome”. After a thorough assessment of this manuscript and to the best of my abilities, I believe that the draft is written well organized and carries fruitful information which might be a good addition in related field of research. However, I have summarized some suggestions to be incorporated which could improve the quality of manuscript.
1. Introduction needs to be summarized.
2. The figures’ quality is very poor and needs to be improved.
3. Figure captions are not informative need to be more informative.
4. Authors used the words “we think that” it need to be replaced with scientific words.
Dear Editor, thank you for letting me review this manuscript entitled “Integrated bioinformatics investigation of novel biomarkers of uterine leiomyosarcoma diagnosis and outcome”. After a thorough assessment of this manuscript and to the best of my abilities, I believe that the draft is written well organized and carries fruitful information which might be a good addition in related field of research. However, I have summarized some suggestions to be incorporated which could improve the quality of manuscript.
1. Introduction needs to be summarized.
2. The figures’ quality is very poor and needs to be improved.
3. Figure captions are not informative need to be more informative.
4. Authors used the words “we think that” it need to be replaced with scientific words.
Author Response
Response to Reviewer 1:
Dear Editor, thank you for letting me review this manuscript entitled “Integrated bioinformatics investigation of novel biomarkers of uterine leiomyosarcoma diagnosis and outcome”. After a thorough assessment of this manuscript and to the best of my abilities, I believe that the draft is written well organized and carries fruitful information which might be a good addition in related field of research. However, I have summarized some suggestions to be incorporated which could improve the quality of manuscript.
Response: We appreciate the time you took to review our manuscript. Thank you for the comments regarding the significance of our draft. Specific comments to the reviewer's remarks:
- Introduction needs to be summarized
Response: We have summarized the Introduction section.
- The figures’ quality is very poor and needs to be improved.
Response: We are sorry for the low quality of the figures. The quality was lost during the conversion from Word to PDF We hope that the revised version of the manuscript provides higher-quality figures.
- Figure captions are not informative need to be more informative.
Response: We have provided more information in Figure We highlighted the changes with green color.
- Authors used the words “we think that” it need to be replaced with scientific words.
Response: We changed "We think" in the Abstract and the Conclusions with "in our " We deleted this phrase elsewhere in the text.
Once again, thank you for your comments. We hope that we revised the manuscript according to your remarks.

Reviewer 2 Report
Thank you for the great opportunity to review this study. The topic addressed is interesting and deserves a constructive discussion. The data and methods may certainly be of use for other uterine leiomyosarcoma studies. However, a major revision of manuscript is needed before it can be accepted for publication.
1. In this manuscript, the authors used multiple previous microarray data to extract DEGs. The data of uterine leiomyoma were used to increase the specificity of marker genes in diagnosing uLMS. However, at present there should be no data in which the origination of uLMS may be from the uLM. For the extraction of DEGs for uLMS, it may be confusing to use uLM data. The reviewer think that the authors should explain the reason for the comparison with uLM in this study clearly.
2. In this manuscript, the authors used the TCGA-SARC database to analyze the relationship between the hub genes and clinical outcomes. However, there is no clear explanation of the background of the cases. Especially, there should have been great influences on the gene expression depending on the stages or the disease progression. The reviewer thinks that it is essential to classify the cases up to the clinical stage or to define the situation of the data. If the authors would like to suggest that the hub genes possibly get the prognosis factors of uLMS, it is necessary that the conditions of sample collection are uniformed for all cases. (For example; the onset of disease.) The reviewer thinks that if the collection points of the samples are at the point in increasing in severity not the onset, these genes possibly be associated with the severity, not prognosis. If the authors would insist the hub genes are the prognosis factors, the stage at the collection time should be the same point, and the tissues collected should be originated from the same places.
3. The current paper only includes a retrospective analysis of OS and DFI in the hub genes. This study lacks experiments to confirm how these genes function in uLMS. The reviewer understands that uLMS cases are very rare. However, the reviewer would recommend including experiments using cell lines.
4. The comparison with Carcinosarcoma is missing from the middle of the report (The comparison was excluded from the analysis after ATRA and PTGER3 were extracted). What are the reasons for the excluding of carcinosarcoma from the analysis?
5. There is a similar report (reference number 71) with this study. In the reference 71, TYMS was noted as a hub gene in PPI in uLMS. The only difference may be that this study includes the analysis from LM data. What is the novelty of this study, in term of this?
6. There are some different words standing for leiomyoma; LM, UFs. They should be uniformed.
Author Response
Thank you for the great opportunity to review this study. The topic addressed is interesting and deserves a constructive discussion. The data and methods may certainly be of use for other uterine leiomyosarcoma studies. However, a major revision of manuscript is needed before it can be accepted for publication.
Response: We want to sincerely thank the reviewer for the detailed revision of our manuscript. Thank you for your time. The questions and remarks raised by the reviewer are important, and we think that we need to explain every point. Also, if the reviewer requires, some of the explanations could be added to the manuscript, if it’s the reviewer’s opinion that this addition would significantly improve the draft.
- In this manuscript, the authors used multiple previous microarray data to extract DEGs. The data of uterine leiomyoma were used to increase the specificity of marker genes in diagnosing uLMS. However, at present there should be no data in which the origination of uLMS may be from the uLM. For the extraction of DEGs for uLMS, it may be confusing to use uLM data. The reviewer think that the authors should explain the reason for the comparison with uLM in this study clearly.
Response: Thank you for this comment. We absolutely agree that we should explain why we compared the data from uterine leiomyomas and uterine leiomyosarcomas. Foremost, the datasets with uterine leiomyosarcomas are relatively sparse. We wanted to obtain as much data as possible, but we included only datasets with more than eight samples for further analysis. Therefore, the first limitation was the number of GSE series. Furthermore, for the homogeneity of the data, we used only processed gene expression by microarray. Moreover, we wanted to present biomarkers that could be potentially used not only for the diagnosis of ULMS but for the distinction between ULMS and ULM, as well as ULMS and other uterine smooth muscle tumors. The DEGs between normal myometrium and ULMS could be used as a starting point in future studies to explore tumor origin from the healthy myometrium. Concerning the reviewer's remark, we think that the inclusion of normal myometrium, ULM, and carcinosarcoma samples, provides enough data to extract the ULMS-specific biomarkers that could, at the same time, be used for diagnosis, prognosis, and explanation of the steps in the oncogenesis process.
- In this manuscript, the authors used the TCGA-SARC database to analyze the relationship between the hub genes and clinical outcomes. However, there is no clear explanation of the background of the cases. Especially, there should have been great influences on the gene expression depending on the stages or the disease progression. The reviewer thinks that it is essential to classify the cases up to the clinical stage or to define the situation of the data. If the authors would like to suggest that the hub genes possibly get the prognosis factors of uLMS, it is necessary that the conditions of sample collection are uniformed for all cases. (For example; the onset of disease.) The reviewer thinks that if the collection points of the samples are at the point in increasing in severity not the onset, these genes possibly be associated with the severity, not prognosis. If the authors would insist the hub genes are the prognosis factors, the stage at the collection time should be the same point, and the tissues collected should be originated from the same places.
Response: We completely agree with the reviewer's remark and we want to thank the reviewer for this question. Since we could not provide a prospective study in a clinical setting, we wanted to evaluate our markers in the TCGA-SARC study. Unfortunately, even after a thorough review of the meta-data, we could not find any information regarding the FIGO stage, TNM stage, or the time of the disease onset. Furthermore, there was, in fact, information regarding the distant metastases at the time of the diagnosis, leiomyosarcoma vascular invasion, and the metastases in the lymph nodes, but the number of samples was small (three and four patients, respectively), so we did not include this data to avoid bias and false discovery. We also further explored this remark. There is, in fact, data from the Memorial Sloan Kettering Cancer Center’s two SARCOMA studies: one from 2010, and the other from 2020. The first study include only somatic status and the fraction of the altered genes, without clinical data. The second study has 165 uterine leiomyosarcoma samples. This study has the following clinical variables: age at which sequencing was reported, cancer type, detailed primary site, FACETS, FGA reported, MSI Score, Overall survival status, primary tumor site, Race category, sample type, Sex, Ethnicity, Purity, and KM plot: overall survival. We wanted to include these samples, but the data regarding gene expression was not permitted. This information was able only upon request from the MSK Cancer Center.
- The current paper only includes a retrospective analysis of OS and DFI in the hub genes. This study lacks experiments to confirm how these genes function in uLMS. The reviewer understands that uLMS cases are very rare. However, the reviewer would recommend including experiments using cell lines.
Response: Thank you for this remark. It is, unfortunately, the main limitation of our study. As we stated in the “Conclusion” section, the expression of DEGs and hub genes should be verified via immunohistochemistry or even genetic studies and correlated with the clinical features of the uLMS patients. Unfortunately, even Human Protein Atlas does not include uterine leiomyosarcoma samples. In our opinion, the results must be validated in the experimental or clinical prospective design. As the reviewer stated, the rarity of the disease prevents us from conducting the mentioned studies at this moment. We wanted to present this study as an introduction for further experiments. Also, we wanted to highlight the hub genes that were significant in the entity of uterine leiomyosarcoma and to validate these results in further experimental and clinical studies. Once again, we completely agree with the reviewer’s remark.
- The comparison with Carcinosarcoma is missing from the middle of the report (The comparison was excluded from the analysis after ATRA and PTGER3 were extracted). What are the reasons for the excluding of carcinosarcoma from the analysis?
Response: We thank the reviewer for this question and we think that it should be further discussed. Table 4 in the Results section presents GO and KEGG pathway enrichment of DEGs between uLMS and UCS. Figure 2B presents the intersected DEGs obtained from all three cohorts. When we extracted ARTX and PTGER3, we did not include these genes in the protein-protein interaction network, since we arbitrarily marked them as significant genes in the uLMS. We also included the dataset with carcinosarcoma samples because, even though the two entities are rare, the highlighted genes in our study could be of potential value in the distinction between uLMS and carcinosarcomas. We included the DEGs from this GEO dataset to obtain more uLMS-specific genes. Since we could not find relevant sources which connect PTGER3 and uLMS, we wrote in detail about ATRX genes in the Discussion section.
- There is a similar report (reference number 71) with this study. In the reference 71, TYMS was noted as a hub gene in PPI in uLMS. The only difference may be that this study includes the analysis from LM data. What is the novelty of this study, in term of this?
Response: The reference number 71 (in a draft before summarizing the introduction section based on remarks from the other review) was: Ren, X.; Tu, C.; Tang, Z.; Ma, R.; Li, Z. Alternative Lengthening of Telomeres Phenotype and Loss of ATRX Expression in Sarcomas (Review). Oncol Lett 2018, 15, 7489–7496, doi:10.3892/ol.2018.8318. does not mention TYMS. Is it possible for reviewer to highlight the name of the mentioned paper?
In terms of novelty, our paper presented the correlation between hub gene’s expression and overall survival in TCGA-SARC patients and further analysied hub gene – transcription factor interaction as well as miRNAs – targed gene network. Finally, our study provided the interaction between TYMS and several different drugs. We could not find similar papers in the literature. Gemcitabine, currently part of the Phase II trial as second-line chemotherapy for uLMs, was one of the drugs in the presented network. In our opinion, the fact that miR-26b-5p (one of the most significant miRNAs in our study), targeted TYMS, has recently been proposed as one of the markers of gemcitabine resistance in patients with bladder cancer, makes this interaction novel in terms of therapy response in uLMS patient, but requires further investigation in expermental and clinical settings. Also, TYMS could potentially be used not only for the initial diagnosis of uLMS, but for the distinction between uterine leiomyomas, and response to therapy in patients which are candidates for the second-line chemotherapy. We sincerely hope that we answered the reviewer question regarding the novelties about the role of TYMS in our paper.
- There are some different words standing for leiomyoma; LM, UFs. They should be uniformed.
Response: Thank you for this remark. We overlooked the abbreviations the reviewer mentioned (mostly in the Materials and methods section). The abbreviation for uterine leiomyomas is now uniform across the whole manuscript – ULM.

Round 2
Reviewer 2 Report
Thank you for the prompt response for the reviewer’s comment. The authors status got clarified by the cover letters for the reviewer. If the authors purpose is for the establishment of the marker for the diagnose of uLMS, their experiment model should be acceptable as they indicate in the cover letter. Additionally, the reviewer totally understand the situation in which the limitation of the meta-data opened. Honestlly, it is certainly that the background of the data should be kept the homogeneity. therefore, it should be better to prepare the exact group even if the numbers are small. But if the nmber is too small, as those were 3, or 4, it may be cause to be confusing. I agree with the authors about the issues.
Also the reviewer was wrongly confused confused about the Q5.
First, the ref# was not 71, that was 87; Fu, Z.; Jiao, Y.; Li, Y.; Ji, B.; Jia, B.; Liu, B. TYMS Presents a Novel Biomarker for Diagnosis and Prognosis in 753 Patients with Pancreatic Cancer. Medicine 2019, 98, e18487, doi:10.1097/MD.0000000000018487.
Secondly, the target was not for uLMS, but Panc C in the paper. The reviewer agree with the novelty which the authors suggested. The reviewer is deeply sorry for the confusion and thanks the authors' kind answer.
Additionally, thank you for the minor fixation in the abbreviation.